

# All-sky photogrammetry techniques to georeference a cloud field

Pierre Crispel[1] and Gregory Roberts[1,2]

[1]CNRM UMR 3589, Météo-France/CNRS, Toulouse, France
[2]Scripps Institution of Oceanography, Center for Atmospheric Sciences and Physical Oceanography, La Jolla, California, USA

*Correspondence to:* pierre.crispel@meteo.fr

**Abstract.**

In this study, we present a novel method of identifying and geolocalizing cloud field elements from a portable all-sky camera stereo network based on the ground and oriented towards zenith. The methodology is mainly based on stereophotogrammetry which is a 3D reconstruction technique based on triangulation from corresponding stereo pixels in rectified images. In cases where clouds are horizontally separated, identifying individual positions is performed with segmentation techniques based on hue filtering and contour detection algorithms. Macroscopic cloud field characteristics such as cloud layer base heights and velocity fields are also deduced. In addition, the methodology is fitted to the context of measurement campaigns which impose simplicity of implementation, auto-calibration, and portability.

Camera internal geometry models are achieved a priori in the laboratory and validated to ensure a certain accuracy in the peripheral parts of the all-sky image. Then, stereophotogrammetry with dense 3D reconstruction is applied with cameras spaced 150 m apart for two validation cases. The first validation case is carried out with cumulus clouds having a cloud base height at 1500 m.agl. The second validation case is carried out with two cloud layers: a cumulus fractus layer with a base height at 1000 m.agl and an altocumulus stratiformis layer with a base height of 2300 m.agl. Velocity fields at cloud base are computed by tracking image rectangular patterns through successive shots. The height uncertainty is estimated by comparison with a Vaïsala CL31 ceilometer located on the site. The uncertainty on the horizontal coordinates and on the velocity field are theoretically quantified by using the experimental uncertainties of the cloud base height and camera orientation. In the first cumulus case, segmentation of the image is performed to identify individuals clouds in the cloud field and determine the horizontal positions of the cloud centers.

## 1 Introduction

Understanding cloud physical mechanisms is essential for understanding climate and meteorological processes. On climate scales, it is recognized that clouds are a major source of incertitude in atmospheric models (IPCC, 2013), whether for the energy balance or water cycle. Yet, many aspects of cloud's life cycle are still not understood by the scientific community (Stevens and Feingold, 2009), hence the need for measurement tools allowing cloud monitoring, particularly in a lagrangian sense.



At present, the instruments most frequently used for remote sensing of clouds from the ground are ceilometers, lidars and cloud radars. Ceilometers and lidars estimate the base height and thickness of several cloud layers. Cloud radars have the same capacities, but are also able to obtain information on the nature of the condensed elements in the cloud (crystals, droplets), as well as their vertical velocities. These ground-based remote sensing instruments are generally oriented towards the zenith

and have a narrow field of view. Cloud radars rotate to reconstruct the cloud field; however the minimum period to complete 360° sweep is a limiting factor for following a cloud field in real time (Borque et al., 2014). Stereophotogrammetry based on a network of all-sky cameras yields three-dimensional information by matching points across stereo images and using triangulation. These techniques provide an inexpensive method to study the three-dimensional organization of a cloud field. The use of all-sky cameras makes it possible to widen the field of view.

Stereophotogrammetry applications for use in meteorology have existed since the beginning of analogue photography (Koppe, 1896), (Bradbury and Fujita, 1968), and more recently digital cameras have been used (Allmen and Kegelmeyer, 1997). In the recent years, several technological advances have been made in camera lenses, image resolution, network communications, computational power and cost reduction. Moreover, major computational improvements have been made in computer vision algorithms, especially in multi-vision reconstruction methods (e.g. *OpenCV* library - Bradski and Kaehler (2008)).

It is now possible to achieve cloud automatic 3D reconstruction by stereophotogrammetry relatively cheaply.

Recent studies on this topic generally use conventional or wide-angle lenses to calculate macroscopic characteristics of a cloud field, such as cloud base heights and cloud layer horizontal velocities. Seiz (2003) uses a pair of wide-angle cameras spaced 800 m apart and pointing to the zenith to calculate the height of the cloud base. The orientation of the cameras is done using the stars. The errors obtained are about 5% for mid-altitude clouds at 4000 m.asl. Hu et al. (2009) use conventional

cameras spaced 1.5 km apart and oriented to mountains to study the three-dimensional organization of orographic convection. The orientation of the cameras is achieved using elements of the landscape. Öktem et al. (2014) are interested in the height of maritime clouds with cameras spaced about 900 m apart. The cameras are oriented towards the horizon. They obtained an error in cloud base height of 2% for low-layer clouds and 8% for cirrocumulus by comparison with lidar measurements. They also calculate an horizontal velocity field that they compare to the data from a radiosonde. In their case, the orientation of the

cameras is achieved by using the position of the sun and the horizon line. In all these previous publications, triangulation is based on the matching of corresponding pixels through the stereo images by manual or automatic methods. In Janeiro et al. (2014), the cloud ceiling information for VFR (Visual Flight Rules) is calculated by matching zenith centered sub-part of the initial stereo images. The authors use low-cost consumer cameras that are oriented towards zenith and spaced about 30 m. The orientation of the cameras is achieved using the stars. For clouds under 1500 m.agl, which are of prime interest for VFR

applications, results at zenith point show good agreement with lidar measurements in single cloud layer situations.

The first study using all-sky cameras in stereophotogrammetry for meteorological purposes is performed by Allmen and Kegelmeyer (1997) to calculate the cloud base height, but temporal synchronization constraints did not allow to obtain usable information. More recently, in order to forecast intra-hour solar irradiance, Nguyen and Kleissl (2014) use their own high resolution all-sky cameras providing very precise equisolid projection. The cameras are spaced 1230 m apart and the authors

use the position of the solar disk to determine orientation. Clouds are filtered in the images with saturation value and cloud base





height is determined by plane-sweeping across the stereo images. The results are compared to ceilometer with 8 hours time series. Residual mean square deviation of 7% for cloud base height at 5000 m.asl is obtained. Three-dimensional reconstruction is also performed and height distribution of triangulated pixels is compared to ceilometer time series showing good agreement. Recently, Beekmans et al. (2016) perform a dense 3D reconstruction from a pair of fisheye lens HD cameras spaced 300 m

apart. The relative orientation of the cameras is estimated using the positions of the stars. This estimation is then refined by an algorithm which automatically matches corresponding stereo pixels. The method is validated by comparison with the data of a ceilometer, a lidar and a cloud radar for a cloud layer of altocumulus stratiformis at about 3000 m. The results show cloud base height relative errors less than 5%. The method is then applied to enable a 3D reconstruction of a developing cumulus mediocris.

In this paper, we use all-sky stereophotogrammetry to perform geolocation of individual elements of a cloud field in order to follow individual clouds in a Lagrangian way, estimate their morphological characteristics and their evolution in real-time. Furthermore, this allows to use cloud geolocation for cloud airborne measurements. For example, in the case of instrumented UAVs, the GPS coordinates of the target cloud may be communicated in real-time to the autopilot. In addition, installation of a camera network for a measurement campaign poses additional challenges. Indeed, it may be difficult, time-consuming, or

sometimes impossible to use landscape elements, or the position of the stars. Therefore the methodology, developed in section 2, is based on the principles of simplicity of implementation, auto-calibration, and portability.

Stereophotogrammetry is based on triangulation: knowing the distance between two cameras, their orientation and the angles of incidence of the optical rays emitted by a physical point, it is possible to find the 3D coordinates of the physical point in a given frame. Thus, several indispensable steps are needed. The calibration of each camera consists in determining how is

mathematically projected an incident optical ray toward the image. This step is carried out in a laboratory using a test pattern. In our case, we use a generic method to perform all-sky camera calibration developed by Scaramuzza et al. (2006). The calibration of the stereo system consists in knowing the distance between the cameras and the relative orientation of each camera. This step is performed once the cameras are installed on the experimental field. In our methodology, positioning and orientation are achieved with GPS, leveling instruments and visual sight, with no obstacles between the two cameras. Precise

relative orientation between the cameras is determined by matching feature points across the stereo images automatically. This is achieved with the SIFT algorithm (Lowe, 2004). The 3D reconstruction step consists in finding for each pixel of the left stereo image, its correspondent in the right stereo image. Three-dimensional information is then calculated by triangulation, involving previously calculated camera internal geometry and orientation. In this work, a dense 3D reconstruction is performed by using a blockmatching method (Szeliski, 2010) on rectified stereo images (undistortion and misalignement corrections).

Additionally, the velocity field is estimated by tracking subparts of the initial image through two successive images and combine this information with the cloud height map. In the case where clouds are sufficiently separated to be considered as identifiable objects, we implement image segmentation for individual cloud georeferencing. We use a color filter to extract the cloud contours of the image and use a segmentation algorithm inspired by Suzuki et al. (1985) to identify cloud objects. Most of the methodology relies on algorithms implemented in open source software libraries: *OcamCalib* (Scaramuzza et al., 2006) for

camera calibration, and *OpenCV* (Bradski and Kaehler, 2008) for the other steps. The accuracy depends on the quality of the





cameras and the algorithms used, as well as on the distance between the cameras, with the following paradox: the greater the distance between the cameras, the better the accuracy but the more difficult pixel matching is.

In section 3, we present the results comparing cloud base heights to traditional methods as well as georeferencing individual cloud elements and calculating the velocity field. The method is applied with cameras spaced 150 m apart, for two validation cases. The first validation case is carried out in the context of a moderately convective situation with isolated cumulus clouds with a cloud base height at 1500 m.agl. The second validation case is carried out in a situation where two cloud layers overlap: a layer of altocumulus stratiformis with a base height of 2300 m.agl and a layer of cumulus fractus with a base height at 1000 m.agl. The height uncertainty is estimated by comparison with a Vaïsala CL31 ceilometer located on the site. The uncertainty on the horizontal coordinates is theoretically quantified by using the experimental uncertainties on the height and uncertainties on the orientation of the cameras. In the cumulus case, a segmentation of the image as well as an estimation of the horizontal positions of the cloud centers is carried out. The results are then discussed in section 4.

## 2    Material and methods

### 2.1    Material

In this work, we use two VIVOTEK FE8391-V network fisheye cameras designed for outdoor video surveillance applications. The focal length is 1.5 mm and the field of view is 180°. The digital sensor is a 12-megapixel CMOS, providing in its full resolution a 2944×2944 px image. The images are transmitted to a computer by a WiFi local network using two directional antennas TP-Link 2.4GHz 24dBi with several hundred meters of range. Horizontal leveling is achieved by the use of a bubble level (accuracy ca. 1°). The respective positions of both cameras in the Earth frame are evaluated by using GPS, and inter camera alignment is achieved with vertical sights on the camera housing.

### 2.2    Camera projection model, calibration and image undistortion

In the camera frame, the projection of an optical ray towards a pixel of the image is generally described by a model which depends on *intrinsic camera parameters*. The camera optical system approaches more or less precisely different types of projections among which the most commonly encountered are the stereographic, equidistant, equisolid, and orthographic projections. In the case of non-scientific cameras, these simple theoretical models are far from sufficient. It is then necessary to use models allowing a better description of the projection by taking imperfections into account (e.g. distortions, offset between optical axis and center of the image, digitization effects). In this article, we use the model proposed by Scaramuzza et al. (2006) to calibrate the cameras. This model was introduced to generically simulate omnidirectional cameras with the property of the single point of view (property generally well approached by a fisheye lens). The intrinsic parameters associated with this model are determined by a calibration step. This calibration is carried out by taking several shots of a flat 2D chessboard pattern. This flexible technique inspired by Zhang (2000) is adapted in the toolbox *OcamCalib* for the Scaramuzza model. One of the advantages of this calibration method is its ease of implementation and its accuracy (e.g., Puig et al., 2012, for a comparative benchmark



between several calibration methods for omnidirectional cameras).

**Pinhole camera model:** The starting point is to consider the simplest camera, that is the *pinhole* camera. It is a box that allows the light rays to pass through a small hole pierced on one side. On the opposite side of the hole, the inverted scene is

projected onto a plate. In order to simplify the way in which this projection is represented, a central symmetry is applied to have a situation in which the image plane and the scene are of the same side with respect to the optical center (Figure 1). The rectangular image plane has an orthonormal coordinate system $(\Omega, \boldsymbol{U}, \boldsymbol{V})$, where $\boldsymbol{U}$ is the horizontal axis of the image and $\boldsymbol{V}$ the vertical axis of the image. The origin $\Omega$ is located at the upper left corner of the image. The camera reference frame is defined by the orthonormal frame $(O, \boldsymbol{X}, \boldsymbol{Y}, \boldsymbol{Z})$, where $\boldsymbol{Z}$ corresponds to the optical axis directed to the observed scene and

$\boldsymbol{X}$ and $\boldsymbol{Y}$ correspond to the $\boldsymbol{U}$ and $\boldsymbol{V}$ axes of the image. The point of intersection of the optical axis with the image is called *principal point*. It does not necessarily coincide with the center of the image, which is especially the case for non-scientific cameras. In this configuration, if $(u', v')$ denotes the centered coordinates of a pixel with respect to the principal point, the projection of a physical point $M(x, y, z)$ is given by the following equation:

$$(u', v') = (f \tan(\phi)\, x/r,\ f \tan(\phi)\, y/r), \tag{1}$$

where $r = \sqrt{x^2 + y^2}$ denotes the distance from the physical point to the optical axis and $\phi = \arctan(r/z)$ denotes the angle of incidence of the optical ray. The parameter $f$ is the pinhole camera focal length (expressed in pixels in the case of a digital camera). Thus, if $(u, v)$ denotes the pixel associated with the $M(x, y, z)$ point in the frame of the image, the projection is defined by:

$$(u, v, 1)^T = \begin{pmatrix} 1 & 0 & u_0 \\ 0 & 1 & v_0 \\ 0 & 0 & 1 \end{pmatrix} (u', v', 1)^T \tag{2}$$

where $(u_0, v_0)$ contains the coordinates of the principal point. We denote $G_{\text{perspective}}^{f, u_0, v_0}$ the projection function of parameters $\{f, u_0, v_0\}$ which maps a physical point $M(x, y, z)$ to a pixel $(u, v)$. The reciprocal projection is denoted by $G_{\text{perspective}}^{-1\, f, u_0, v_0}$. It maps an optical ray $\{\lambda(x, y, 1),\ \lambda \in \mathbb{R}\}$ to a pixel $(u, v)$.

**Omnidirectional Scaramuzza model:** Under the axisymmetric assumption, and if $r'$ denotes the distance between $(u, v)$ and

the principal point, equation (Eq. 1) can be generalized to:

$$(u', v') = (r'(\phi)\, x/r,\ r'(\phi)\, y/r). \tag{3}$$

The distance $r'$ in pixels depends on $\phi$ and characterizes the radial distortions. These distortions are preponderant in a fisheye lens. This is the reason why the function $r'(\phi)$ is called *representation function* of the fisheye lens. In the Scaramuzza model, this function is implicitly defined by the relation $\tan \phi = -r'/p(r')$ where $p(r')$ is a polynomial function $p(r') = a_0 + a_1 r' +$

$\ldots + a_n r'^n$. The tangential distortions are taken into account linearly by an additional correction step (parameters $c$, $d$ and $e$).





Thus, if $(u, v)$ denotes the pixel associated with the $(x, y, z)$ point in the frame of the image, the projection is defined by:

$$(u, v, 1)^T = \mathbf{M}(u', v', 1)^T = \begin{pmatrix} 1 & e & u_0 \\ d & c & v_0 \\ 0 & 0 & 1 \end{pmatrix} (u', v', 1)^T. \tag{4}$$

We denote $G_{\text{fisheye}}^{\mathbf{M}, a_0, \ldots, a_n}$ the fisheye projection function of intrinsic parameters $\{\mathbf{M}, a_0, \ldots, a_n\}$.

**Camera calibration method:** The camera calibration determines the camera intrinsic parameters $\{\mathbf{M}, a_0, \ldots, a_n\}$. To do this, we use $N$ shots of a chessboard with $K_1 \times K_2$ corners (intersections between black and white tiles - Figure 3). We denote by $\mathcal{R}_{\text{chessboard}}$ a coordinate system such that the origin is located on one of these corners, and that the horizontal axes coincide with the chessboard lines. For each shot $i$, and for each chessboard corner $(x_j, y_j, 0)_{\mathcal{R}_{\text{chessboard}}}^T$, we have the relation:

$$(u_{ij}, v_{ij})^T = G_{\text{fisheye}}^{M, a_0, \ldots, a_n} \left( \mathbf{R}_i (x_j, y_j, 0)_{\mathcal{R}_{\text{chessboard}}}^T + \boldsymbol{T}_i \right) \quad i = 1, \ldots, N \quad j = 1, \ldots, K_1 \times K_2, \tag{5}$$

where $(u_{ij}, v_{ij})$ denotes corners positions on the image, $\mathbf{R}_i$ the rotation from the camera frame to $\mathcal{R}_{\text{chessboard}}$ and $\boldsymbol{T}_i$ the translation between the optical center of the camera and the origin of $\mathcal{R}_{\text{chessboard}}$. The calibration is based on the following steps using the toolbox *OcamCalib*:

1. For each shot $i$, corners are automatically detected in the image using the intensity gradient specific signal and the pattern of the board (Figure 3 bottom). This process gives $(u_{ij}, v_{ij})$ values.

2. In the nonlinear system (Eq. 5), the values of $(u_{ij}, v_{ij})$ and $(x_j, y_j)$ are known and the system is overdetermined for sufficiently large values of $N$ and $K_1 \times K_2$. Parameters $\{\mathbf{M}, a_0, \ldots, a_n, \{\mathbf{R}_i, \boldsymbol{T}_i, \forall i = 1 \ldots N\}\}$ are determined by using a Marquardt-Levenberg method.

**Undistortion:** In order to produce undistorted images, the scene is reprojected according to a conventional centered perspective projection of focal length $f$. During this reprojection, we move from a circular fisheye image to a square image of size $N_{\text{px}} \times$

$N_{\text{px}}$. The intensity of each pixel of the undistorted image is calculated according to the relationship:

$$\text{RGB}_{\text{undistorted}} (u_{\text{undistorted}}, v_{\text{undistorted}}) = \text{RGB}_{\text{fisheye}} \left( G_{\text{fisheye}}^{\mathbf{M}, a_0, \ldots, a_n} \circ G_{\text{perspective}}^{-1 \; f, N_{\text{px}}/2, N_{\text{px}}/2} (u_{\text{undistorted}}, v_{\text{undistorted}}) \right). \tag{6}$$

In this transformation, the peripheral areas of the image are weakly resolved. These fields are filled by interpolation, hence the blur effect. Note that the values of $f$ and $N_{\text{px}}$ can be freely chosen. The field of view of undistorted images $\text{FOV}_{\text{undistorted}} = 2 \arctan \frac{N_{\text{px}}}{2f}$ will depend on these values. The smaller the value of $f$, the larger the field of view, but the more interpolated

areas occupy an important part of the image.

### 2.3    Orientation, stereo calibration and rectification

At the end of the previous step, we are able to produce two undistorted stereo images. They are square images of the same size $N_{\text{px}} \times N_{\text{px}}$, for which the center of the image and the principal point coincide, and which would have been taken by two



pinhole cameras with the same focal length $f$. The next step consists in orienting them with respect to each other as accurately as possible. To achieve this, Hu et al. (2009) use landscape features and Öktem et al. (2014) use the horizon line. Seiz (2003) and Beekmans et al. (2016) use the positions of the stars that allows to determine the orientation of each camera. In addition, they add an algorithmic correction step based on SIFT stereo pixel matching algorithm (Lowe, 2004). In our work, we develop

a visual orientation method assuming that there is no visual obstacle between the two cameras. Like Beekmans et al. (2016), this initial orientation is refined by an algorithmic step.

**Orientation and stereo calibration:** The cameras optical axis are oriented towards the zenith. The image planes are at the same altitude, and the horizontal axes of the undistorted images are aligned. This theoretical orientation of the all-sky stereo

system is called *frontally aligned* (Figure 4). From the GPS coordinates, the distance $b = O_1 O_2$ between the cameras and the angle of deviation from the North $\beta = \widehat{N O_1 O_2}$ are calculated with Haversine formulas. The orientation of the equipment requires an additional calibration step, which is algorithmic and determines the precise relative orientation of camera frames. This procedure is usually referred to as *stereo calibration* and consists of calculating the components of the relative rotation $\mathbf{R}$ and the relative translation $T = \boldsymbol{O_1 O_2}$ between camera frames such as $(x,y,z)^T_{\mathcal{R}_1} = \mathbf{R}(x,y,z)^T_{\mathcal{R}_2} + \boldsymbol{T}$.

Stereo calibration is based on the concepts and theorems of *epipolar geometry*. In particular, in the case of pinhole cameras with the same focal length $f$, it exists a constant matrix $3 \times 3$ of rank 2 denoted $\mathbf{E}$ and called *essential matrix*. This matrix only depends on $\mathbf{R}$ and $\boldsymbol{T}$ and verifies the following constraint:

$$(u_2'^M, v_2'^M, 1)^T \mathbf{E} (u_1'^M, v_1'^M, 1) = 0, \tag{7}$$

for all pixels $(u_1^M, v_1^M)$ from the left stereo image, and $(u_2^M, v_2^M)$ from the right stereo image representing the same physical

point $M$. We use the following stereo calibration methodology:

1. From the undistorted stereo images, retrieve a set of stereo matching pixels with the SIFT algorithm (Lowe, 2004).

2. Using the pairings of step 1, solve the overdetermined system (7) whose unknowns are the coefficients of the matrix $\mathbf{E}$. We use a Least Median of Squares (LMEDS) regression, which avoids being affected by outliers. The matrix $\mathbf{E}$ is determined to within a scalar factor.

3. Calculate $\mathbf{R}$ and $\boldsymbol{T}$. For this purpose, the following equations are used:

$$[\mathbf{T}_\times]^2 = -\mathbf{E}\mathbf{E}^T \quad \text{with } [\mathbf{T}_\times] = \begin{pmatrix} 0 & -T_z & T_y \\ T_z & 0 & -T_x \\ -T_y & T_x & 0 \end{pmatrix}, \tag{8}$$

that give two opposite solutions $\boldsymbol{T}_+$ et $\boldsymbol{T}_-$ and

$$\mathbf{R} = \frac{1}{\|\boldsymbol{T}\|^2} \left( [\boldsymbol{E_2} \times \boldsymbol{E_3} \ \ \boldsymbol{E_3} \times \boldsymbol{E_1} \ \ \boldsymbol{E_1} \times \boldsymbol{E_2}] \pm [\mathbf{T}_\times]\mathbf{E} \right), \tag{9}$$

where $\boldsymbol{E_k}$ denotes the $k$-th column of the $\mathbf{E}$ matrix. The uniqueness of the solution is obtained by requiring the scene to

be located in front of the cameras as well as the constraint $\|\boldsymbol{T}\| = b$.





4. Corrective rotations $\mathbf{R}_1$ and $\mathbf{R}_2$ are defined by using $\mathbf{R}$ and $\boldsymbol{T}$ such that:

$$\mathbf{R}_1 = \mathbf{R}_{\mathbf{rect}}^{-1} \quad \text{and} \quad \mathbf{R}_2 = \mathbf{R}_{\mathbf{rect}}^{-1}\mathbf{R}. \tag{10}$$

where $\mathbf{R}_{\mathrm{rect}} = (\boldsymbol{e_1}, \boldsymbol{e_2}, \boldsymbol{e_3})$ is a rotation matrix such as $\boldsymbol{e_1}$ is oriented in the same direction of $\boldsymbol{T}$, and $\boldsymbol{e_2}$ is orthogonal to $\boldsymbol{e_1}$ and to the left camera optical axis .

5. Consistency step: $\mathbf{R} \simeq \mathrm{Id}$ and $\boldsymbol{T} \simeq (b,0,0)^T$. This validation is necessary because the algorithm is sensitive to the stereo pixel matching quality and can, in some cases, converge towards an incoherent minimum. In practice, we minimize $\|\boldsymbol{T} - (b,0,0)^T\|^2 + \|\mathbf{R} - \mathrm{Id}\|^2$ by varying subsets of matching pixels used in step 2 ($\mathbf{E}$ matrix computation).

**Rectification:** We use $\mathbf{R}_1$ and $\mathbf{R}_2$ to produce undistorted rectified images; that is, the images that would have been produced by perfectly aligned pinhole cameras. These images are produced from all-sky original images by the following transformation:

$$\mathrm{RGB}_{\mathrm{rectified}}^{\mathrm{CAM1,2}} \left(u_{\mathrm{rectified}}, v_{\mathrm{rectified}}\right) = \mathrm{RGB}_{\mathrm{fisheye}}^{\mathrm{CAM\ 1,2}} \left(G_{\mathrm{fisheye}}^{\mathrm{intrinsic\ params\ CAM\ 1,2}} \circ G_{\mathrm{perspective}}^{-1\ f,N_{\mathrm{px}}/2,N_{\mathrm{px}}/2} \circ \mathbf{R}_{1,2} \left(u_{\mathrm{rectified}}, v_{\mathrm{rectified}}, 1\right)\right). \tag{11}$$

## 2.4 Three-dimensional reconstruction

Three-dimensional reconstruction is obtained by triangulation from two pixels $(u_1^M, v_1^M)$ and $(u_2^M, v_2^M)$ which are known to represent the same physical point $M$. Indeed, knowing the projection functions of each camera, their relative orientations, and the distance between the cameras, it is possible to estimate the point of intersection of the optical rays in a given reference frame. Working directly with the rectified images make this calculation easier because we have a simple theoretical standard situation: identical pinhole images in a frontally aligned orientation (Figure 4 right). In this case, two matching pixels are located on the same row in the image matrices (i.e. $v_1^M = v_2^M$). Then, the coordinates $(x_M, y_M, h_M)$ in the rectified frame of the left camera are given by:

$$h_M = \frac{f\,b}{u_2^M - u_1^M} = \frac{f\,b}{\delta^M}, \quad x_M = h_M \frac{u_1'^M}{f}, \quad y_M = h_M \frac{v_1'^M}{f}. \tag{12}$$

where $\delta^M = u_2^M - u_1^M$ is called *disparity* and is linearly related to $h$ through the baseline distance between the cameras $b$, and the focal distance $f$.

In addition, a dense 3D reconstruction of the observed scene assumes that one is able to generate a dense matching of corresponding pixels across the stereo images. This is called the *dense stereo matching problem*. In the case of rectified images, this problem is greatly simplified by the fact that $v_1^M = v_2^M$, and thus, becomes a one-dimensional problem. In this case, a very common method is the block matching algorithm (Szeliski, 2010), which relies on finding maximum correlations between neighborhoods of pixels across the stereo images. This algorithm is implemented in the OpenCV library (Bradski and Kaehler, 2008), and is able to describe finely the variations of altitude. However, it generates noise/speckles in weakly textured image part which is a disadvantage for the type of objects that we consider (clouds, blue sky background, sun). To avoid this effect, we use several techniques:

– adjusting algorithm parameters: height detection range can be limited during the pixel matching process by setting minimum and maximum bounds for cloud height detection ; window correlation size is adjusted to prevent speckles.

(c) Author(s) 2017. CC BY 4.0 License.




- smoothing the signal by reducing the size of the image while taking advantage of the subpixel resolution of the algorithm.

- using blue sky filtering: we process the altitude map by filtering the blue sky areas. We use image conversion in the HSV color management system (Hue, Saturation, Value). The hue values ranging from 170° to 280° (from cyan to violet) are filtered.

## 2.5 Velocity field

The estimation of the cloud field horizontal velocity is carried out by using two successive rectified images $I^{t_1}$ and $I^{t_2}$ coming from the same camera. Using cross-correlation techniques, the displacement of the cloud field from one image to another is evaluated in pixel units. This displacement on the image is converted into velocity by using the previously calculated height map. In practice, the initial image is divided into rectangular blocks $I^{t_1}_{k_1,k_2}$ indexed by the subscripts $k_{1,2}$ (Figure 5). The median of heights $h_{k_1,k_2}$ is assigned to these blocks based on the cloud height map. The translation in number of pixels of each block through two successive shots is denoted by $\mathbf{\Delta}_{k_1,k_2}$. It is related to the block mean horizontal velocity $\left( v^x_{k_1,k_2}, v^y_{k_1,k_2} \right)$ by:

$$v^x_{k_1,k_2} = \frac{h_{k_1,k_2}}{f} \frac{\Delta^u_{k_1,k_2}}{\Delta t}, \quad v^y_{k_1,k_2} = \frac{h_{k_1,k_2}}{f} \frac{\Delta^v_{k_1,k_2}}{\Delta t}, \tag{13}$$

where $\Delta t = t_1 - t_2$ is the time between two shots. Calculating $\mathbf{\Delta}_{k_1,k_2}$ is to determine the position of a $I^{t_1}_{k_1,k_2}$ template in the $I^{t_2}$ image. This generic computer vision problem is called *template matching*. A method developed by Lewis (1995) and based on the Normal Cross Correlation index allows to perform this search with a low algorithmic cost in simple cases (no rotation, no scaling). This algorithm is available in the OpenCV library (Bradski and Kaehler, 2008).

Note that the technique used here is similar to that used by Janeiro et al. (2014), which evaluates the displacement of a single block centered on the principal point through two images. In our case, the approach is multiblock, which generates dispersion but makes it possible to estimate the velocities of multiple cloud layers.

## 2.6 Segmentation and cloud identification

Segmentation techniques are used in computer vision problems to identify objects in an image. In our case, the main interest of this technique is to identify and georeference individual clouds when the situation allows it (e.g. cumulus cloud field). Segmentation is achieved with the following steps:

1. production of a binarized image from blue sky filter (Section 2.4).

2. contour detection and segmentation from binarized the image: we use a contour finding algorithm implemented in OpenCV library and inspired by Suzuki et al. (1985).

3. filtering non-significant/noisy contours: we eliminate contours with a low inside area, and with a low number of inner triangulated pixels.



4. filtering sun: we use a threshold on altitude to remove the sun.

Each segmented region contains pixels that have been triangulated in the 3D reconstruction process. This allows to assign $(x, y, z)$ coordinates for each triangulated pixel. In order to avoid outliers the center of each segmented cloud, and the cloud base height is estimated with:

$$x_{\text{center}} = \frac{q_5(\underline{\text{x}}) + q_{95}(\underline{\text{x}})}{2}, \quad y_{\text{center}} = \frac{q_5(\underline{\text{y}}) + q_{95}(\underline{\text{y}})}{2}, \quad z_{\text{cloud base}} = q_{10}(\underline{\text{z}}) \tag{14}$$

where $\underline{\text{x}}$, $\underline{\text{y}}$, $\underline{\text{z}}$ corresponds to coordinates of all triangulated pixels within the segmented region. The notation $q_r(\underline{\text{x}})$ (resp. $\underline{\text{y}}$ and $\underline{\text{z}}$) denotes the $r^{\text{th}}$ quantile of $x$ values (resp. $y$ and $z$) within the segmented region.

## 2.7 Uncertainty estimation

Theoretically, in a frontally aligned pinhole stereo system, the uncertainty on height $\sigma_h$ can be related to the uncertainties on the position of corresponding pixels $(u_1, v)$, $(u_2, v)$, given by the sensitivity equation:

$$\sigma_h = \sigma_{|u_1 - u_2|} \frac{h^2}{fb} = \sigma_\delta \frac{h^2}{fb}. \tag{15}$$

where $\sigma_{|\delta|} = \sigma_{|u_1 - u_2|}$ represents the uncertainty on disparity (Section 2.4). This equation shows that uncertainty decreases linearly as the baseline distance $b$ increases, until a distance here the quality of the stereo-pixel matching degrades. On the other hand, $\sigma_h$ quadratically increases with increasing heights.

In a practical way, the uncertainty related to the 3D reconstruction of the cloud field in the Earth's frame has several components: camera resolution, intrinsic projection/calibration model, position and orientation of the cameras/stereo calibration, and pixel matching. We quantify the overall uncertainty on cloud base height experimentally. In this work, we use a Vaïsala CL31 ceilometer, collocated with the all-sky stereo system, as the reference instrument. It provides information by measuring the cloud base height at the zenith, and identifies up to three cloud layers. Several aspects must be identified before comparing ceilometer and all-sky stereo system results:

1. there is spatial inter-cloud and intra-cloud variability of the cloud base height.

2. all-sky stereo system computes heights coming from the base as well as the sides of the clouds.

3. ceilometer provides a point value at zenith, while the cameras provide a spatial map of the heights.

4. all-sky stereo system can recover multiple cloud heights only if it can see them.

Several methodologies can be used to compare all-sky spatial data to ceilometer temporal data. A comparison of height measurements at zenith when the picture is taken allows to estimate uncertainty on height $\sigma_h$, although this method is limited because it does not represent the uncertainty on the peripheral parts of the image. Another way is to compare the height frequency histograms obtained by the all-sky stereo system (heights calculated for a scene) with the distribution of the heights obtained by the ceilometer (centered time series). The distribution peaks represent the representative height of the cloud base





for a given cloud layer. The thickness associated with these peaks is due to the above-mentioned uncertainties and cloud base variability. The error is estimated by comparing the peak positions and the standard deviations of the distributions around these peaks.

In the Earth frame, the uncertainty on $(x, y)$ position can be deducted from uncertainties on height $\sigma_h$, polar angle $\sigma_\phi$,
and azimuthal angle $\sigma_\theta$. Indeed, in spherical coordinates we have $x = \rho \cos\theta \sin\phi$, $y = \rho \sin\theta \sin\phi$, $h = \rho \cos\phi$. By denoting $r = \sqrt{x^2 + y^2} = h \tan\phi$, the ground projected distance, we obtain $x = h \cos\theta \tan\phi$ and $y = h \sin\theta \tan\phi$, such as:

$$\sigma_x^2 = (\cos\theta \tan\phi)^2 \sigma_h^2 + (h \sin\theta \tan\phi)^2 \sigma_\theta^2 + (h \cos\theta \cos^{-2}\phi)^2 \sigma_\phi^2, \tag{16}$$

$$\sigma_y^2 = (\sin\theta \tan\phi)^2 \sigma_h^2 + (h \cos\theta \tan\phi)^2 \sigma_\theta^2 + (h \sin\theta \cos^{-2}\phi)^2 \sigma_\phi^2, \tag{17}$$

$$\sigma_r^2 = \tan^2\phi \, \sigma_h^2 + h^2 \cos^{-4}\phi \, \sigma_\phi^2. \tag{18}$$

The angle uncertainties are mainly related to the orientation of the cameras: initial orientation (GPS position and visual sighting) and algorithmic correction in the rectification process (Section 2.3). In our study, the estimation of $\sigma_h$ is calculated experimentally as mentioned above. The corrective rotations provided by the rectification algorithm in different configurations allows to estimate $\sigma_\theta$ and $\sigma_\phi$, providing $\sigma_\theta = \sigma_\phi = 2°$.

## 3 Results

### 3.1 Camera calibration

The *OcamCalib* toolbox allows computing both camera intrinsic parameters following Scaramuzza's model (Section 2.2). This calculation is made from multiple images of a chessboard (30 shots in our case). Validation of the calibration is based on two indicators: the first indicator is the mean re-projection error which is the mean root square of the difference in pixel units between corner positions as estimated through the calibrated model and those initially detected. The second indicator measures
the maximum value of this difference. These values (in pixel units) are compared with the size of the images produced by the cameras: 2944 px $\times$ 2944 px. Dispersion can also be quantified by comparing the representation functions determined for both cameras. The calibration parameters are listed in Table 1. Figure 6 (left panel) shows the $r'_{\text{CAM1}}(\phi)$ representation function of camera 1, as calculated by the calibration method. This function, which characterizes the projection, is compared with typical fisheye parametric models. The difference between the representation functions of the cameras 1 and 2 is shown in Figure 6
(right panel). The calculation of sensitivity $d\phi/dr'$ allows to estimate the uncertainty on the angle of incidence $\phi$. In our case, this uncertainty varies as a function of $\phi$ between 0.06°/px and 0.07°/px. Finally, Figure 7 illustrates the dispersion of the reprojection errors for each corner and for each shot.

The result of this calibration shows that both all-sky cameras obtain an intermediate projection between the equidistant and equisolid projections (Figure 6 left), with the difference increasing significantly beyond an incident angle of 50° (10 px
deviation / angular error of 0.65° ). This shows that the use of a precise calibration model and method is needed if one wishes to use the peripheral parts of the all-sky image. The difference between the representation functions of the camera 1 and the



camera 2 (Figure 6 right) shows that the fisheye projections are almost identical up to an angle of incidence of 70° (2 px deviation), which is an indicator of validity of the calibration. This uncertainty increases significantly beyond angles greater than 80°. The dispersion of the reprojection error is small (Table 1, Figure 7) with an average reprojection error less than 1 px (i.e, 0.065°) and 7 px (i.e, 0.5°) maximum deviation .

## 3.2 Georeferencing results

### 3.2.1 Validation cases

We apply the methodology described in section 2 for two types of cloud fields. In both cases, measurements are made on the Météo-France site in Toulouse, France. The baseline distance between the cameras is given by $b \simeq 150$ m. The first validation case is carried out in the context of a weakly convective mid-afternoon situation (July 2016). Clouds are cumulus humilis, mediocris and congestus. The cloudiness is around 4 octas. In this case, the cloud base height is around 1500 m.agl, which implies a ratio $h/b \simeq 10$. The second validation case occurred around noon, highlighting the detection of multiple cloud layers (June 2016). The clouds are cumulus fractus with cloud base at 1000 m.agl and altocumulus stratiformis with bases at 2300 m.agl. The cloudiness is about 6 octats. In this case, we have a ratio $h/b \leq 15$. The context of each test case is summarized in Table 2.

### 3.2.2 Cloud height map

For each validation case, we repeat the same procedure three times at intervals of 15 min.

1. Capture and undistortion of the fisheye images (Section 2.2).

2. Stereo pixel matching and stereo calibration (Section 2.3).

3. Rectification of the undistorted images (Section 2.3).

4. 3D reconstruction and calculation of the height map (Section 2.4).

5. Comparison with a $\pm$ 15 min time series from the ceilometer (Section 2.7).

Note that in operational situations, the stereo calibration (step 2) does not need to be performed before each shot if the material stays in place. Since we quantify the error associated with the entire methodology, step 2 is re-executed for each shot. In step 4, smoothing and filtering techniques to avoid speckles in non-textured zones are implemented (Section 2.4). In particular, min/max threshold on heights is set to $h \in [450 \text{ m}, \ 4000 \text{ m}]$ and a blue-sky filter is implemented. The parameters for image undistorsion, rectification, 3D reconstruction and segmentation are given in Table 3.

We compare the distributions of the heights obtained with the all-sky stereo system to the ceilometer. The results obtained for the first and second case are presented in Figures 8 and 9, respectively, with images spaced 15 min apart. On those panels, top row represents the undistorted and rectified images of the left camera at each time interval. The middle row represents





cloud height maps. The bottom row represents the distributions of the calculated heights (blue histograms) compared with the ceilometer distributions (red histograms). A comparison of these distributions is summarized in Figure 10. For images where a cloud intersects the zenith direction, a ponctual comparison between ceilometer and all-sky system heights is given with the histograms.

The results of the cumulus case show that, for cameras spaced 150 m apart and a cloud base height of about 1500 m.agl, the cloud base height distributions obtained with the all-sky stereo system are similar to the ceilometer. The maximum offset on the distribution peak is about $\pm 150$ m, which is $\sigma_h/h \simeq 10\%$ for a $h/b \simeq 10$ ratio. Around these peak values, the data show a standard deviation $\delta_h \simeq 100$ m, which is similar to the ceilometer. As expected, instantaneous comparison at zenith gives better accuracy results with a measurement difference up to 50 m (i.e. $\sigma_h/h \simeq 3\%$).

The results of the second validation case (altocumulus/multi-layer) show that all-sky camera network can identify multiple cloud layers. In this case, the offset between distribution peaks is 20 m for the lower cumulus fractus cloud layer ($h = 1000$ m.agl). For the layer at 2300 m.agl, the offset on the distribution peak varies between 60 m (2nd image) and 350 m (3rd image). In this case, where $h/b = 15$, the ratio $\sigma_h/h \simeq 15\%$. As previously stated, standard deviations obtained by the cameras and the ceilometer are similar around the peak values, varying between 100 m and 200 m. For this case, instantaneous comparison at

zenith gives a measurement difference up to 100 m for the 2300 m layer (i.e. $\sigma_h/h \simeq 5\%$).

From these experiments, using peak distribution offsets, we note that $\sigma_h/h \simeq 0.01\,h/b$ can be considered as a general rule for the height measurement uncertainty when using our methodology. The stereo calibration step is most likely responsible for the observed shifts. As we have explained in section 2.3, this step is sensitive to the quality of the pixel matching performed by the SIFT method. This is illustrated by Figure 11 showing variability of the height distribution with different stereo calibrations

in the altocumulus case. According to $\sigma_h/h$ relationship, sensitivity on the stereo calibration step increases when ratio $h/b$ increases. Indeed, in this example, the peak corresponding to the low-layer cumulus fractus clouds ($\simeq 1000$ m) is barely impacted by the stereo calibration step.

### 3.2.3  Horizontal georeferecing results, velocity map and segmentation

The horizontal georeferencing and velocity results obtained for the cumulus and altocumulus/multi-layer cases are shown in

Figures 12 and 13, respectively. For each figure, we show: the left camera rectified image and its associated velocity map (top figures), the 3D point cloud projection on the left camera $x - y$ horizontal plane, and uncertainty on position $\sigma_r$ (bottom figures). This uncertainty is estimated using the equation (18) with $\sigma_\phi = 2°$, and $\sigma_h = 0.01\,h^2/b$ according to the experimental results presented in the previous section.

The results show the ability of the all-sky stereo system to retrieve information on cloud field spatial organization. As

expected, position uncertainty increases with altitude and angle of incidence of the cloud. For the cumulus case, this uncertainty is about 120 m for a cloud located at a ground distance of 1 km, 250 m for a cloud located at 2 km and 450 m for a cloud located at 3 km. The estimated velocity is 14 km/h with a mean direction of wind of 255°.

For the multi-layer/altocumulus case, the uncertainty is 180 m for a cloud at 1 km, 330 m for a cloud at 2 km and 500 m for a cloud at 3 km. The velocity results show that the all-sky stereo system is able to estimate the velocities of different cloud





layers. In this case, the estimated average velocity is 16 km/h for the 1000 m layer and 30 km/h for the 2300 m layer, with respective mean directions of 205° and 230°.

We note that the uncertainty on cloud layer velocity is related to $h$ following Eq.13, and is between 10% and 15% in the cases studied.

In the cumulus mediocris case, as we have separated cumulus clouds on the images, the situation allows to go further and implement the segmentation algorithm (Section 2.6). Results are shown in Figure 14. We show the cloud height map, as well as the segmented image with the estimated positions of cloud centers (red dots and cloud identification number). The estimated cloud positions are listed in Table 4. The estimated individual cloud base heights are compared with the $\pm$ 15 min ceilometer time series. In our case, we find that the all-sky camera network allows to identify clouds as individual objects. The estimated

cloud base heights agree well with the ceilometer.

## 4   Discussion and future work

The results obtained under the configuration described in this study are relevant for macroscopically characterizing a cloud field up to 2500 m altitude, as well as cloud targeting applications by instrumented UAVs. Yet, for precise measurements: morphological parameters of a cloud (width, vertical extension and variation over time), precise geolocation (e.g. measurements

near the base, top, or edges of the cloud), the all-sky camera network must be configured to ensure a certain accuracy.

In addition to optimizing the baseline distance between the cameras, several strategies can be explored to improve the accuracy of all-sky camera system. A first strategy is to work on the robustness of the orientation step. Relative orientation accuracy between stereo cameras plays an important role in the image rectification process (Section 2.3). Indeed, relative orientation has an impact on 3D reconstruction accuracy through pixel matching hit score, and uncertainty on disparity, as

shown with (Eq. 12) and (Eq. 15), and experimentally in Figure 11. Moreover, it is important to ensure that cameras are correctly oriented in the Earth's frame for accurate geolocalization, as shown in (Eq. 18).

In previous studies, the camera orientation is based on identified elements of the landscape, such as stars, trees, buildings and horizon lines. This consideration of external elements is adapted to the context of a fixed installation of a camera system, but becomes less suitable in the context of a measurement campaign in which the all-sky camera network must be mobile and

rapidly operational. The technique used here to initially orient the camera network is based on: GPS for positioning in the Earth frame, leveling for horizontal adjustment, and vertical sights on the camera housing for inter camera alignment, which is a priori less accurate than using landmarks or stars to establish the orientation. Improving the initial orientation accuracy can be accomplished using laser sighting or the use of successive images of a GPS-equipped balloon or UAV loitering in the field-of-view of the cameras. In addition, the relative orientation between camera-pairs can be refined by the stereo calibration

algorithm using a time series of several pairs of images, instead of an instantaneous snap shot of a single pair of images. In addition, improved accuracy can also be achieved by organizing a network of several cameras (Heinrichs et al., 2007). For example, the arrangement of the cameras on the ground can be used to increase the number of triangulations of the same object





(e.g. square arrangement with four cameras). Inter-camera spacing can also be organized to accommodate different cloud layers (e.g., closely-spaced cameras for low clouds and farther apart for high altitude clouds).

For dense stereo matching, the block-matching algorithm (Szeliski, 2010) yields correct results even in weakly textured areas, provided that smoothing and filtering techniques are implemented (Section 2.4). However, smoothing step impacts

accuracy when reconstructing cloud edges. Block-matching algorithm is a standard method and it would be useful to carry out a comparative study of the results given by dense matching methods developed recently. This field of research is very active and there is a dedicated benchmark on-line platform described in Scharstein and Szeliski (2002). One of the objectives of a future study would be to use this benchmark to identify and implement methods capable of accurately characterizing low-textured cloud zones, as well as edges.

In terms of image segmentation (e.g. identification of individual clouds), and geolocation, the methods and results presented in this article provide an overview of computer vision techniques to estimate individual cloud positions and their characteristics in a shallow cumulus cloud-field. Segmentation based on contour detection of neighboring pixels makes it possible to isolate individual clouds. The cloud segmentation approach used in this study, works well for distinguishable clouds on the image, but its performance is less reliable if this is not the case. The cloud segmentation method can be refined by taking into account

the altitude map for more complex cloud fields where different clouds overlap on the image (e.g. multiple cloud layers, higher cloudiness, or deep convection). We see in Fig. 9 that the reconstruction algorithm determines low cumulus fractus edges within overlapping higher cloud layer. For a stratiform cloud layer with high cloudiness and less contrast, the segmentation approach would be modified to discern macroscopic differences in the cloud structure. Nonetheless, as mentioned in the previous paragraph, reconstructing accurate edges in situations where low textured objects overlap remains a challenging task

in the computer vision field. The uncertainty with respect to geo-localization of an individual cloud center position is directly related to uncertainty estimation on height (Section 2.7).

Finally, the use of photogrammetry techniques associated with segmentation opens the way to the characterization of other parameters of interest to the atmospheric science, such as the width of the cloud base and the vertical extension of the clouds, as shown by Beekmans et al. (2016). In addition, segmentation makes it possible to track individual clouds through successive

images and follow the evolution of the cloud life cycle.

*Acknowledgements.* This study has been performed within the framework of the Skyscanner project supported by the STAE foundation and the Micro-Aerial Vehicle Research Center. We particularly thank Frédéric Murguet at Météo-France, and the CNRM/GMEI/MNPCA team for technical and organizational support. We also thank Samuel Lauda for preliminary studies on this project, and Simon Lacroix (CNRS/LAAS) for support on computer vision techniques.





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





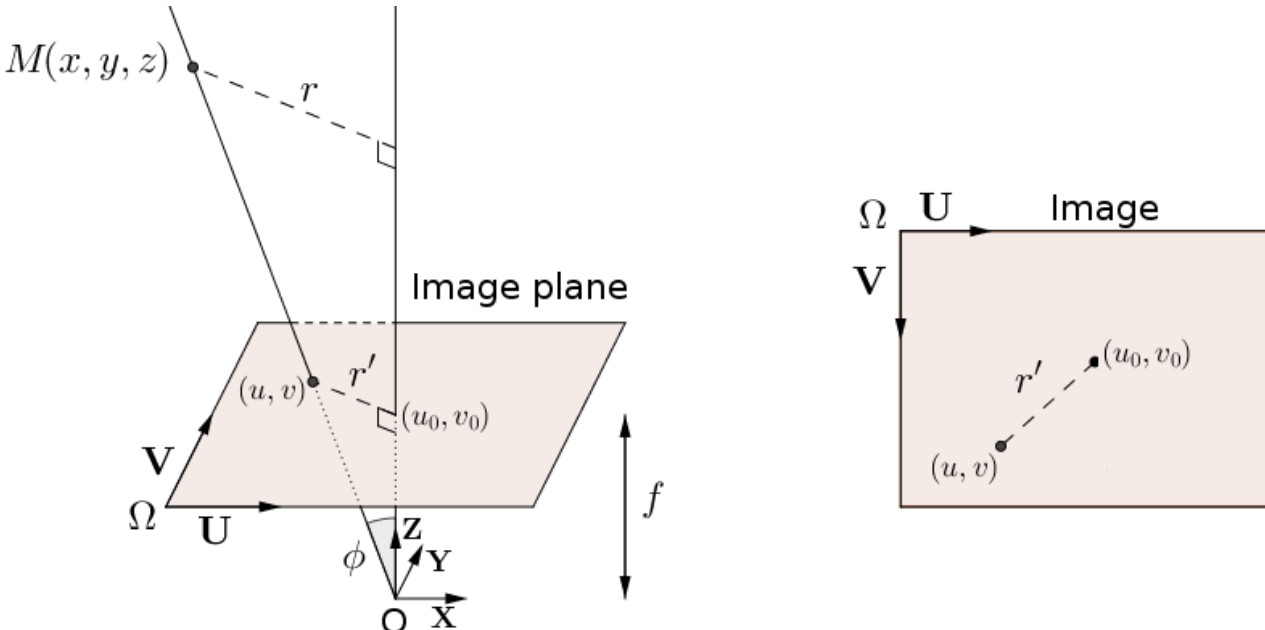

**Figure 1.** Pinhole projection. Physical point $M$ is projected on $(u, v)$ on the image-plane $(\Omega, \boldsymbol{U}, \boldsymbol{V})$. Camera coordinate system is defined by axis $\boldsymbol{X}$ and $\boldsymbol{Y}$ which are colinear to $\boldsymbol{U}$ and $\boldsymbol{V}$, and by axis $\boldsymbol{Z}$ which is the *optical axis*. *Principal point* $(u_0, v_0)$ is the projection of the *optical center* $O$ on the image. The radial projected distance on the image is denoted $r'$.

**Table 1.** OcamCalib calibration results for cameras 1 and 2. Parameters are described in equations (Eq. 3) and (Eq. 4) in section 2.2.

|  |  | principal point | | radial distortion parameters | | | | |
| --- | --- | --- | --- | --- | --- | --- | --- | --- |
|  | Image size | $u_0$ | $v_0$ | $a_0$ | $a_1$ | $a_2$ | $a_3$ | $a_4$ |
| Camera 1 | 2944×2944 | 1467.6 | 1468.0 | −980.6 | 0 | 3.9853e-4 | −1.0973e-7 | 1.0861e-10 |
| Camera 2 | 2944×2944 | 1452.5 | 1452.8 | −982.4 | 0 | 3.5975e-4 | −2.3627e-8 | 6.2340e-11 |

|  | affine distortion parameters | | | Re-projection errors | |
| --- | --- | --- | --- | --- | --- |
|  | $c$ | $d$ | $e$ | RMS | Max |
| Camera 1 | 0.9999 | 3.12e-4 | −7.55e-4 | 0.7 px | 5.3 px |
| Camera 2 | 0.9999 | 5.68e-4 | −9.44e-4 | 1.0 px | 7.5px |



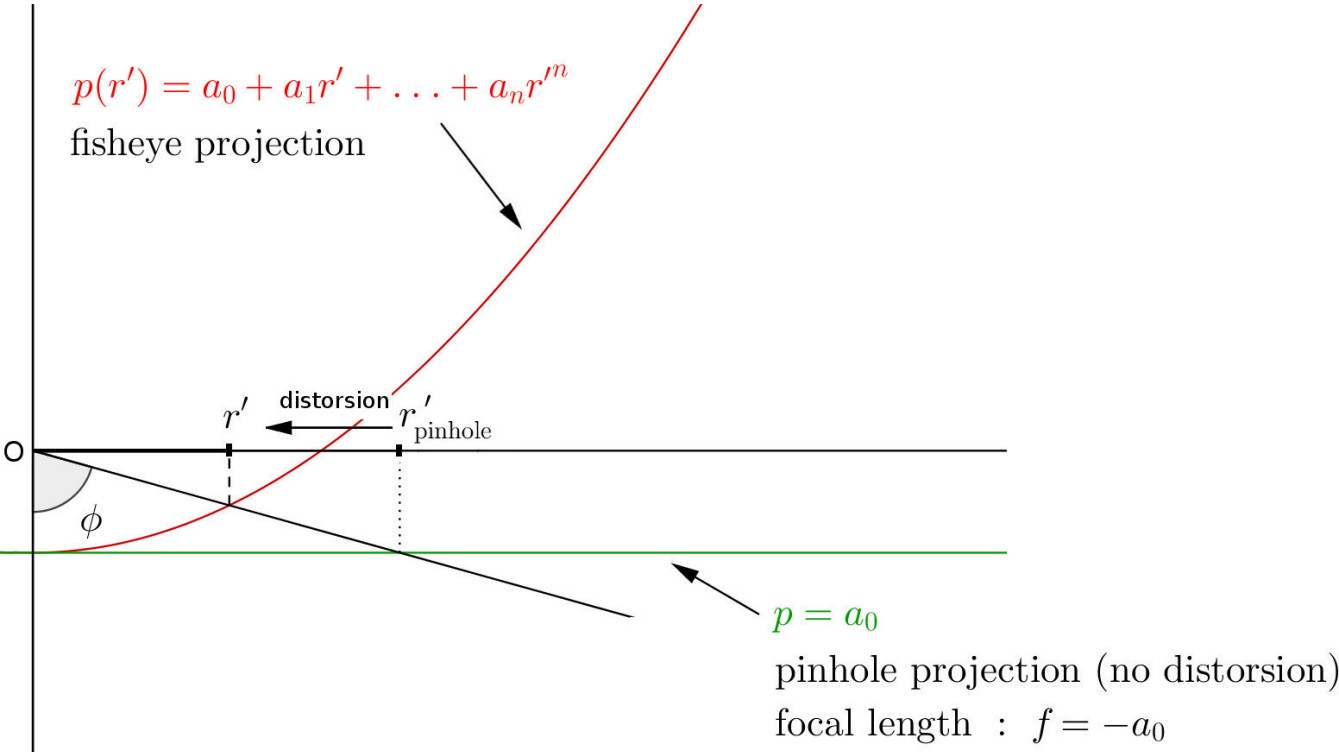

**Figure 2.** Radial distorsion modelization in Scaramuzza et al. (2006) for omnidirectional cameras. Incident angle $\phi$ and projected radial distance $r'$ are related by $\tan\phi = -r'/p(r')$. The polynomial function $p$ is represented by the red curve. The case where there is no distorsion (i.e. pinhole projection $r' = f\tan\phi$) corresponds to a constant polynomial function $p = a_0$ represented by the green line.

**Table 2.** Validation cases description

|  | Validation case 1 | Validation case 2 |
|---|---|---|
| Place | Toulouse - Météo France | |
| Date (UTC) | 2016-07-08 13:55 | 2016-06-16 10:00 |
| Shots | 3 shots every 15 minutes | |
| Baseline distance between cameras, $b$ | 147 m $\pm$ 3 m | |
| Type of clouds | cumulus humilis and mediocris | cumulus fractus and altocumulus stratifromis |
| Mean cloudiness | 50% (4 octas) | 75% (6 octas) |
| Mean cloud base height | 1500 m.agl | 1000 m.agl and 2300 m.agl |
| Ceilometer | Frequency : 1 min, Start/End time of temporal serie = $\pm$ 15 min before and after camera shot | |





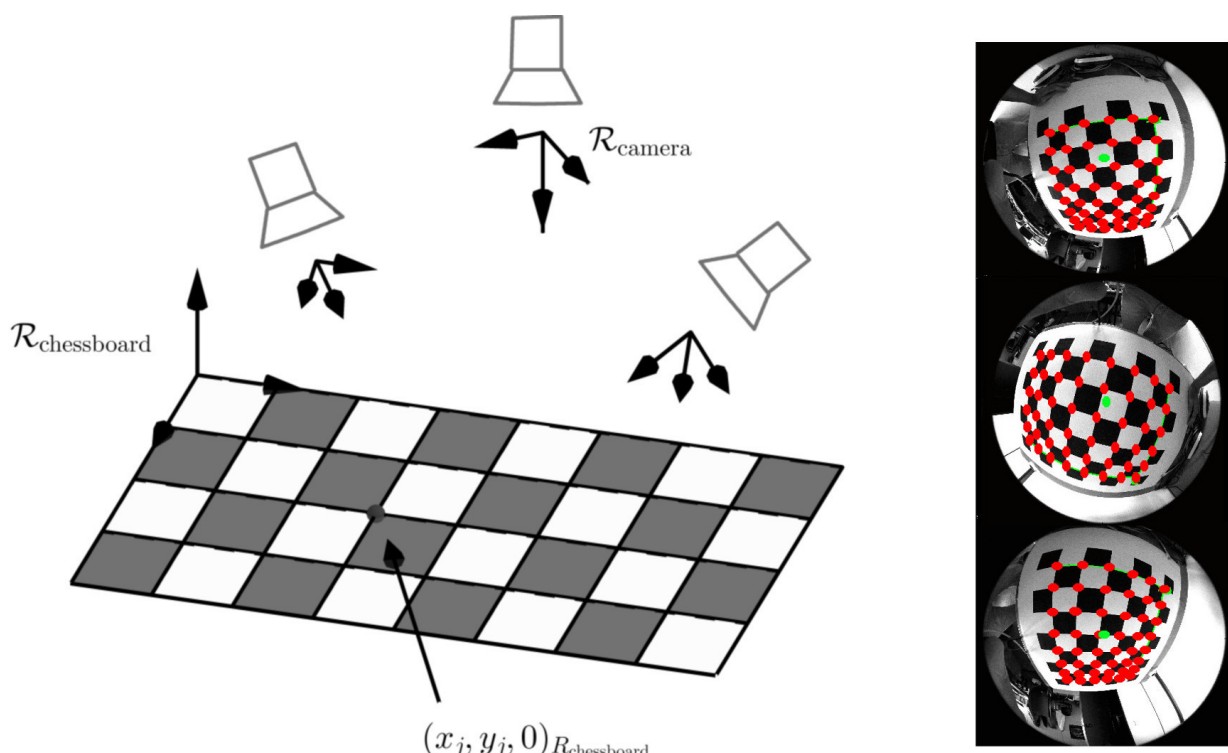

**Figure 3.** Calibration procedure by multiple views of the same chessboard. The procedure is automatized by using an algorithmic corner detection. The camera projection function is estimated with the *OcamCalib* toolbox following Scaramuzza et al. (2006) modelization.

**Table 3.** Algorithm parameters

| | |
|---|---|
| Undistorsion / rectification algorithms | Size of undistorted and rectified images: $N_{\mathrm{px}} = 2944$ |
| | Field of view of undistorted/rectified images 136° |
| Block matching algorithm parameters | size reduction of images : $512 \times 512$, max height = 4000 m, min height = 400 m, |
| | correlation window size = 11 px, subpixel scale : 1/16 |
| Segmentation / Significative contours thresholds | $\frac{\text{inside contour area (px}^2)}{\text{total image area}} > \frac{1}{1000}$, $\frac{\text{Nb of triangulated pixels in contour}}{\text{Nb of pixel in contour}} > 75\%$. |




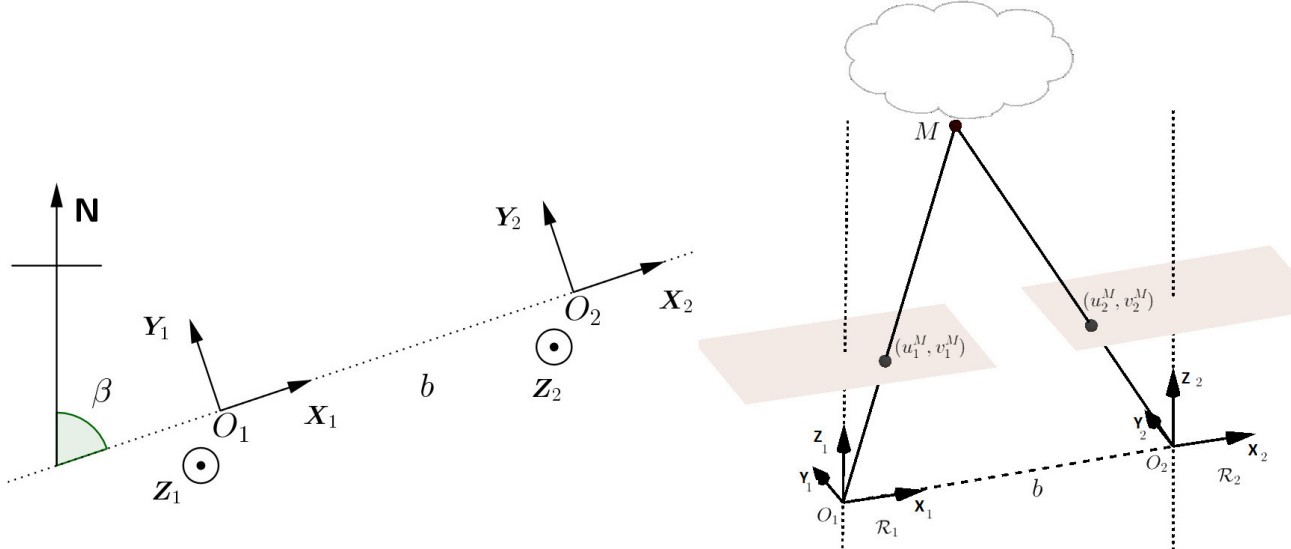

**Figure 4.** Ideal camera configuration. Camera coordinate systems are frontally aligned with optical axis $z_{1,2}$ oriented towards zenith. Optical centers $O_{1,2}$ are in the same altitude plane. The baseline distance is denoted $b$ and North bearing of $O_1O_2$ axis is denoted $\beta$. In this ideal configuration, assuming that we have identical pinhole centered cameras, corresponding pixels $(u_1^M, v_1^M)$ and $(u_2^M, v_2^M)$ are row aligned on the imagers (i.e. $v_1^M = v_2^M$).

**Table 4.** Segmentation and geolocalization results.

| Cloud Id | Estimated cloud base height (m.agl) $\pm$ 10% | Position $(x, y)$ of cloud centers in the left rectified coordinate system | $r$ | $\sigma_r$ |
|---|---|---|---|---|
| 3 | 1440 | (-2.69 km, 1.75 km) | 3.21 km | $\pm$ 350 m |
| 5 | 1670 | (2.41 km, 1.55 km) | 2.87 km | $\pm$290 m |
| 6 | 1420 | (-1.83 km, 1.46 km) | 2.34 km | $\pm$260 m |
| 7 | 1450 | (-1.80 km, -0.23 km) | 1.81 km | $\pm$170 m |
| 9 | 1430 | (-0.68 km, -1.00 km) | 1.21 km | $\pm$120 m |
| 10 | 1450 | (1.35 km, -1.57 km) | 2.10 km | $\pm$210 m |
| 12 | 1640 | (-0.23 km, -2.89 km) | 2.90 km | $\pm$290 m |

Ceilometer cloud base heights measured during a 30 minutes time series:

1420 - 1450 - 1530 - 1350 - 1560 - 1550 - 1630 - 1620 m.agl



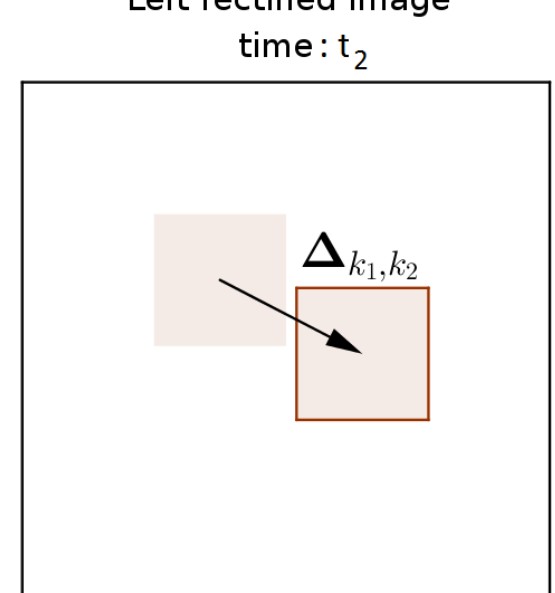

**Figure 5.** Multiblock tracking algorithm for cloud field velocity estimation. For each block $I_{k_1,k_2}$, velocity vector is computed by using the displacement vector $\boldsymbol{\Delta}_{k_1,k_2}$ expressed in pixels and the median altitude $h_{k_1,k_2}$. Displacement vector is computed by using the Lewis (1995)'s matching template algorithm. Computations are based on two successive rectified images: in our case we use the left rectified image at times $t_1$ and $t_2$.





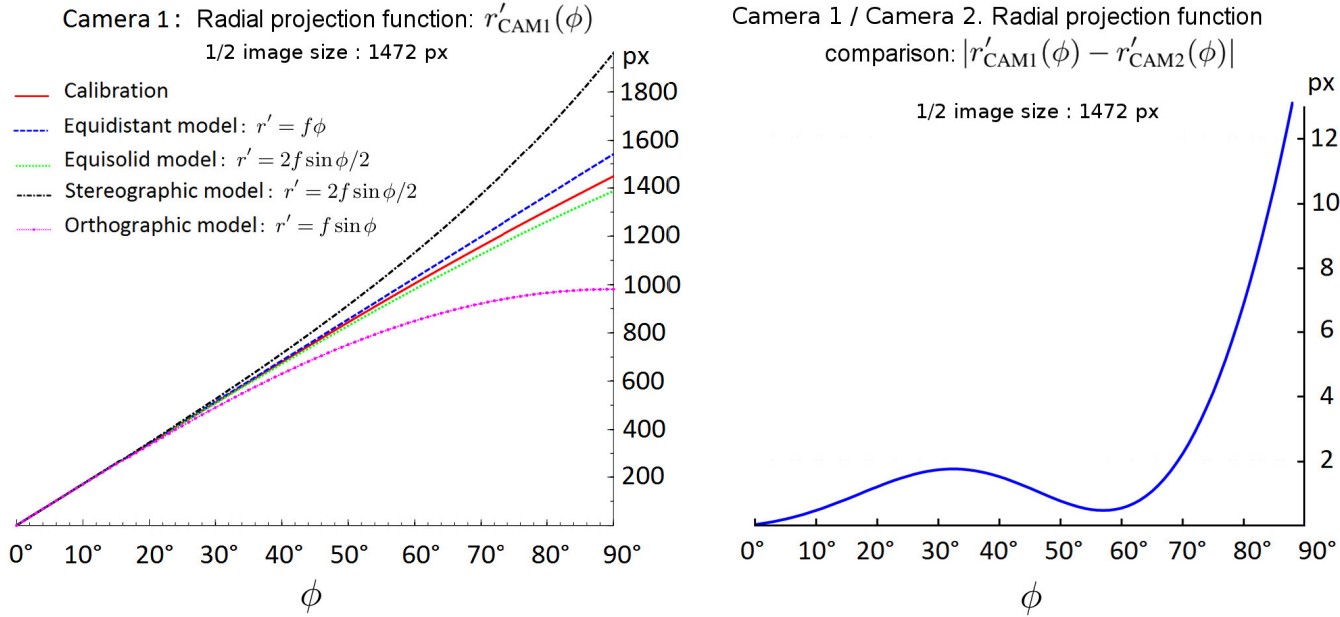

**Figure 6.** Left figure : radial projected distance $r'$ as a function of incident angle $\phi$ for VIVOTEK camera 1. Function $r'(\phi)$ is called *representation function* as it characterize the projection. It is compared to the mostly used fisheye parametric representation functions setted with $-a_0$ value for $f$. Right figure : difference in pixels between representation functions of camera 1 and camera 2, as a function of incidence angle $\phi$.





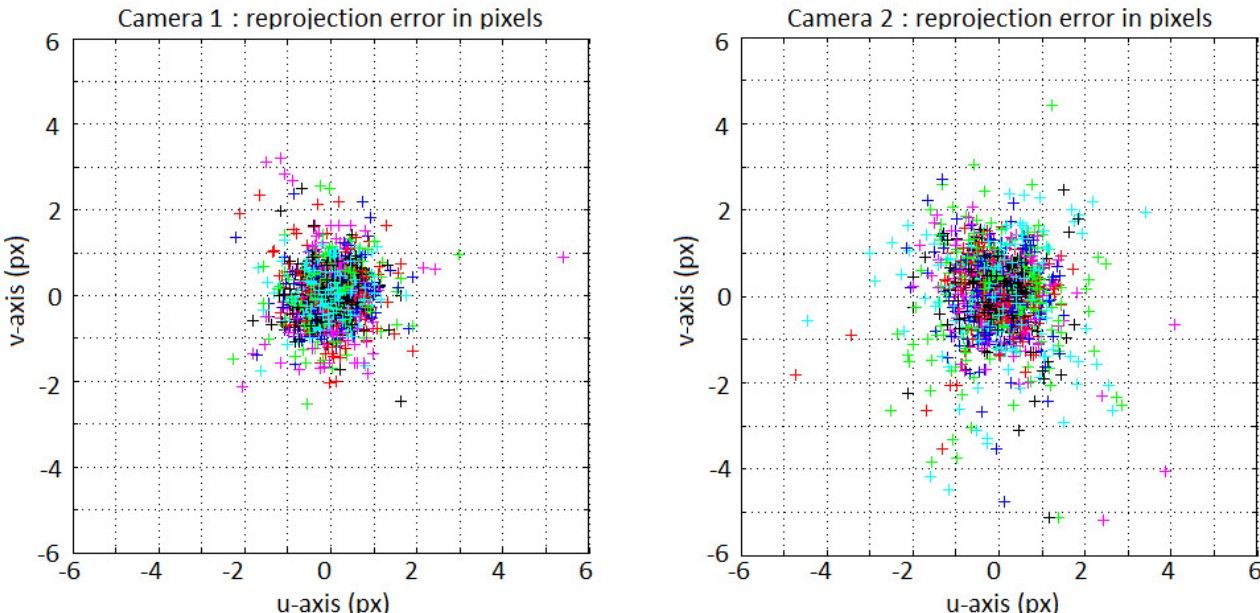

**Figure 7.** For each view and each chessboard corner (which represents an amount of $30 \times 48$ points), difference between corner position on the image, and corner position computed by re-projection, using *OcamCalib* calibration results.





**Figure 8.** Cumulus validation case - Height map and distribution for 3 shots evenly spaced of 15 min. Top row represents the left camera images. Middle row represents the associated height map computed by the stereovision system. Bottom row represents the frequency histogram of heights computed by the stereovision system (blue diagram - 100 m bins). This distribution is compared to the ceilometer frequency histogram (red curve - 100 m bins).

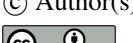



**Figure 9.** Altocumulus/multilayer validation case - Height map and distribution for 3 shots evenly spaced of 15 min. Top row represents the left camera images. Middle row represents the associated height map computed by the stereovision system. Bottom row represents the frequency histogram of heights computed by the stereovision system (blue diagram - 100 m bins). This distribution is compared to the ceilometer frequency histogram (red curve - 100 m bins).





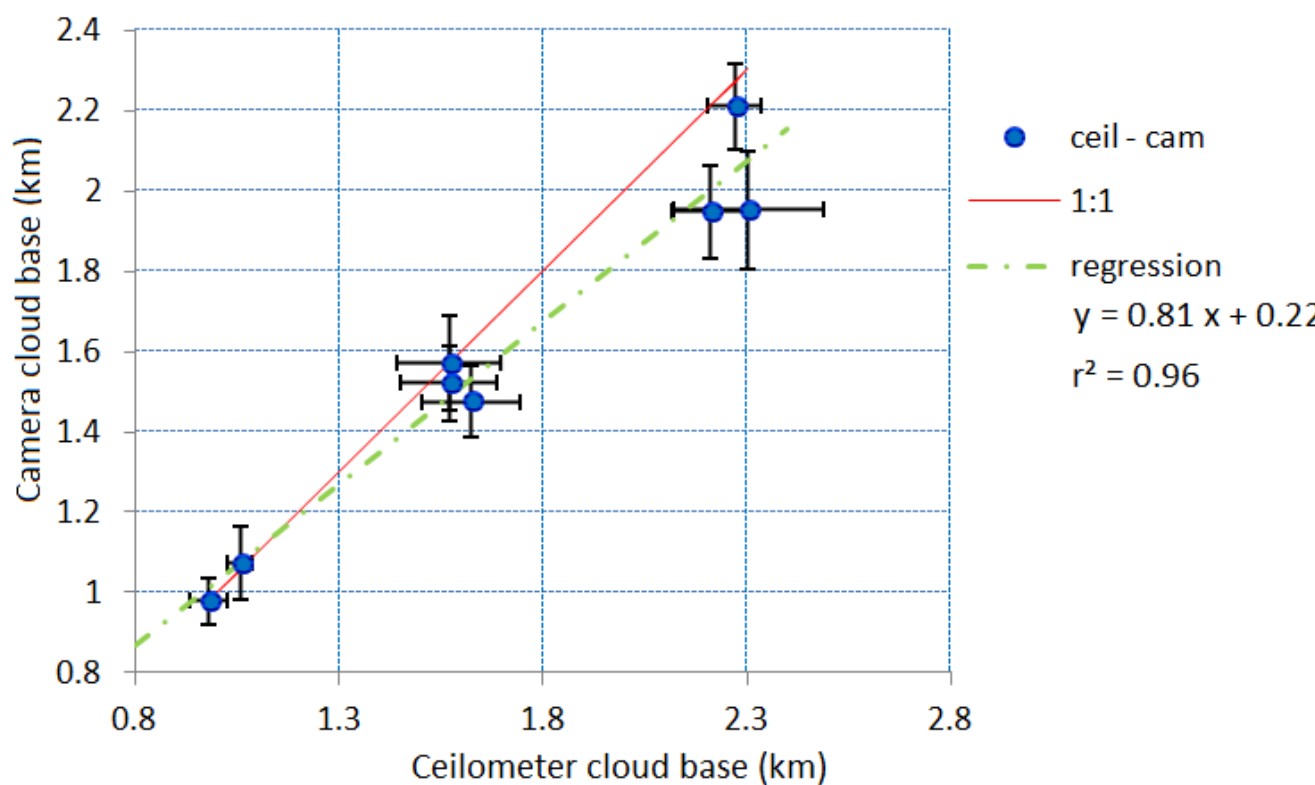

**Figure 10.** Comparison of mean cloud base height results obtained by the camera stereovision system and mean cloud base height results obtained by the ceilometer (blue points). The red 1:1 line corresponds to the reference plot. Linear regression for the ceilometer-camera plot is shown in green dashed line.



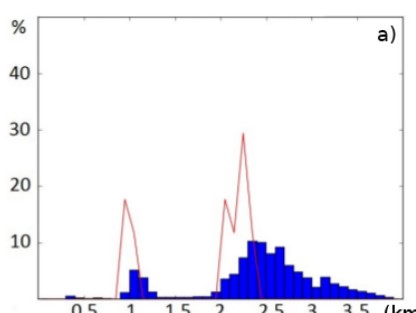
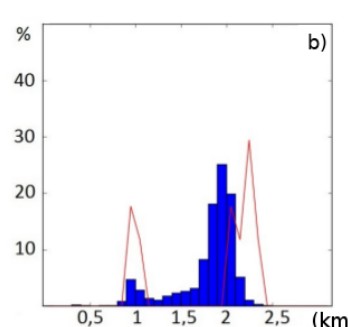
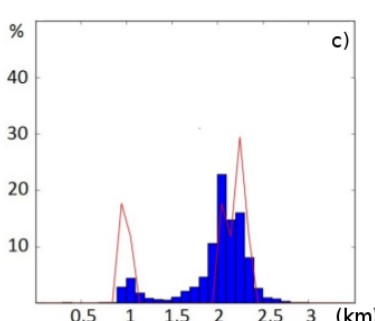

**Figure 11.** Sensitivity of the stereo calibration step illustrated by the first shot in the altocumulus validation case. For this shot, we represent the height frequency histogram obtained with : a) no stereo calibration, b) stereo calibration parameters obtained with the first shot pair of images, c) stereo calibration parameters obtained with the second shot pair of images.



**Figure 12.** Cumulus case - Top figures : rectified image (left) with estimated wind speed and direction (right). Bottom figures : triangulated points projected on $x - y$ left camera plane with altitude colormap (left), and with $r$-incertitude colormap (right).


**Figure 13.** Altocumulus/multilayer case - Top figures : rectified image (left) with estimated wind spped and direction (right). Bottom figures : triangulated points projected on $x - y$ left camera plane with altitude colormap (left), and with $r$-incertitude colormap (right).





**Figure 14.** Top: Undistorted and rectified left image with associated height map. Bottom/left: Contours produced by blue filtering segmentation on left rectified image. Bottom/right: Segmented image with cloud identification number and estimated position of center of cloud base (red dots). Altitude filter: 4000 m.agl