# Peer review of "All-sky photogrammetry techniques to georeference a cloud field"

_Atmospheric Measurement Techniques, 2017_

## Referee Comment (RC1) · Anonymous Referee #2 · 26 Sep 2017

This article presents a method for geolocalizing clouds and extracting cloud base heights and horizontal velocity fields by use of a pair of all-sky cameras at zenith angle. The idea of using stereo cameras in cloud field observation is not novel, but the authors of this work eliminates the external calibration overhead by setting cameras at zenith angle and using leveling instruments. External calibration introduces extra constraints for stereo setup, hence eliminating- or simplifying- this step enables portability. Another contribution introduced in this work is the estimation of velocity fields by cloud segmentation followed by block search/matching. This method enables automatic extraction of vertical velocity fields at the cloud height.

The authors discuss the benefits and limitations of their setup and methods in comparison with ceilometer observations. An additional limitation of the proposed setup might

be in extracting vertical profiles of clouds, since cameras cannot see through clouds and all sky cameras work at zenith angle. The authors may consider discussing the impact of this limitation in investigating cloud's life cycle.

Page 2, line 24, "an" needs to be "a" Table 3, replace "significative" with "significant" Figure 8/9, center/rightmost panel in the bottom row, are the numbers missing, what is "//"?
* * *

---

## Referee Comment (RC2) · Anonymous Referee #1 · 28 Sep 2017

General Comments:

The submitted manuscript deals with the application of stereo-photogrammetry to image data obtained from a pair of all-sky cameras and aims at the geolocalization, i.e. the determination of geographically meaningful coordinates, and identification of individual shallow cumulus clouds. While the exact geographic position of a cloud is not relevant for parameters such as cloud area, base height or cloud evolution, it is relevant for in-situ measurements, for example via the mentioned UAV-based sensors, or remote sensing with cloud radars or satellites. In general, geolocalization is implicitly given when operating a stationary camera pair, e.g. by using stars or landmarks as orientation, as is done in related studies. However, the problem is much more significant in the case of a mobile camera pair, where such sources of orientatioin are difficult if

not impossible to include. The article describes in detail the full procedure, starting with the applied camera calibration step, stereo-rectification, matching and 3-D reconstruction, including a theoretical and empirical uncertainty analysis. The article should be published after dealing with the following remarks.

Specific Comments:

- In section 1, page 3 line 24, the authors write that their approach to derive the relative orientation is based on the "visual sight, with no obstacles between the cameras". On the other hand, in section 2.3, page 7 line 12, the authors describe that the relative orientation is "achieved with vertical sights on the camera housing". I suppose that this aims at retrieving an initial relative orientation estimation before refining it using the SIFT features. Given the fact, that this is a central step in this study and a major difference to other studies that use stars or landmarks, this procedure should be described in more detail and clarity. Is the visual sight used to eliminate the remaining degrees of freedom in the relative orientation? Maybe an additional figure or an extension of Fig. 4 might be useful here. In addition, the two figures in Fig. 4 could be merged quite easily, i think.

- Regarding the segmentation step described in section 2.6, page 9 line 29, the authors mention that the technique is applied "when the situation allows it(e.g. cumulus cloud field)". Altough the authors mention on page 15 line 16, that the current method has some limitations in case of cloud overlap and propose to use cloud height values for compensation, the images shown in Fig. 8 (t, t+15 min) suggest that even in a cumulus case, a clear separation of clouds using such contour-based methods alone might be difficult, resulting in a merged contour rather than two separate ones. Maybe the authors could write an additional sentence about this problem in section 2.6 already. This does not touch the presented case studies of the segmetation, which are fine.

- In section 2.2, page 6 line 22, the authors mention the blur effect in the peripheral regions of the rectified image. To my understanding, the blur effect results from the

mapping of a given image region of the fisheye image onto a larger projection area in the rectified image. I find the term "interpolation" misleading, because the rectification itself is performed in reverse, from given rectified image coordinates for which the corresponding coordinates in the fisheye image are computed. Since such an image mapping generally never hits the center of a pixel in the target image exactly, an interpolation is applied as a normal procedure (e.g bilinear).

- There seems to be a minor error in equation 7, where the two vectors are written as row vectors, but have the transposition applied as in the normal co-linearity equation, which generally assumes column vectors. As far as i can see, the rest of the article uses column vectors (e.g. page 7 line 17). Hence, the row vector on the right side of the essential matrix must be a column vector and vice versa.

- In section 2.3, page 8 line 11, a constraint for the matrix R is introduced, which enforces R to be the identity. In general, the matrix R is to be computed in the relative orientation estimation procedure and should not be the identity matrix. Of course, the relative orientation of the cameras in a frontally aligned pose should be the identity, but not the matrix R, which is to be computed.

- In section 2.4, page 9 line 5, the authors describe a method to filter out outlier in the reconstruction by introducing a lower and upper limit of a valid height. This might work well for zenith regions, but for larger incidence angles mismatches introduce a larger error in the horizontal location rather than the vertical, even of the lower and upper limits for the height value are satisfied. In other words, a depth value error moves from a vertical error to a more and more horizontal error as the incidence angle grows larger. This limitation should be mentioned here.

Technical Corrections:

- Page 3 line 19+20: This sentence should be reformulated. For example: "The calibration of each camera encompasses a mathematical description of the projection of an incident optical ray onto the image."

- Page 10 line 2: "contour detection and segmentation using the binarized image: .."

- Page 10 line 18: ".. ., until a distance -w-here the quality..."

- Page 13 line 3: "undistorsion" - > "undistortion"

- Page 13 line 14: Should the delta_h be sigma_h?
* * *

---

## Author Comment (AC1) · 24 Nov 2017

We thank the reviewer for his constructive comments and respond to him individually in the following text.

**Reviewer #1:** In section 1, page 3 line 24, the authors write that their approach to derive the relative orientation is based on the "visual sight, with no obstacles between the cameras". On the other hand, in section 2.3, page 7 line 12, the authors describe that the relative orientation is "achieved with vertical sights on the camera housing". I suppose that this aims at retrieving an initial relative orientation estimation before refining it using the SIFT features. Given the fact, that this is a central step in this study and a major difference to other studies that use stars or landmarks, this procedure should be described in more detail and clarity. Is the visual sight used to eliminate the remaining degrees of freedom in the relative orientation? Maybe an additional figure or an extension of Fig.4 might be useful here. In addition, the two figures in Fig. 4 could be merged quite easily, i think.

→We agree. The following figures illustrating the installation have been added.

[Figure]

**Figure 1**: Top - VIVOTEK FE8391-V fisheye camera and installation structure. Bottom - Vertical sights on the camera housing allows visual inter-camera alignment in the horizontal plane.

[Figure]

→Left and right parts of Figure 4 are merged in one figure (now Figure 5)

**Figure 5**: Ideal camera configuration. Camera coordinate systems are frontally aligned with optical axis $z_{1;2}$ oriented towards zenith. Optical centers $O_{1,2}$ are in the same altitude plane. The baseline distance is denoted $b$ and North bearing of $O_{1,2}$ axis is denoted $\beta$. In this ideal configuration, assuming that we have identical pinhole centered cameras, corresponding pixels $(u^M_1, v^M_1)$ and $(u^M_2, v^M_2)$ are row aligned on the imagers (i.e. $v^M_1 = v^M_2$).

→The following sentence (Line 12 p.7):

The orientation of the equipment requires an additional calibration step, which is algorithmic and determines the precise relative orientation of camera frames.

has been replaced by:

Initial orientation of the cameras is previously described in Figure 1 and gives an orientation of the cameras close to the ideal frontally aligned orientation. However, this procedure is not sufficient to perform an accurate 3D reconstruction which needs row alignment of corresponding stereo pixels in the stereo images (see Figure 5). A refining algorithmic step to calculate the precise relative orientation of the cameras and consequently rectify the stereo images is then required.

**Reviewer #1:** Regarding the segmentation step described in section 2.6, page 9 line 29, the authors mention that the technique is applied "when the situation allows it(e.g. cumulus cloud field)". Although the authors mention on page 15 line 16, that the current method has some limitations in case of cloud overlap and propose to use cloud height values for compensation, the images shown in Fig. 8 (t, t+15 min) suggest that even in a cumulus case, a clear separation of clouds using such contour-based methods alone might be difficult, resulting in a merged contour rather than two separate ones. Maybe the authors could write an additional sentence about this problem in section 2.6 already. This does not touch the presented case studies of the segmentation, which are fine.

→ We develop the paragraph in line 22 p. 9:

[...] In our case, the main interest of this technique is to identify and georeference individual clouds when the situation allows it. The method that we present here is a contour-based method involving blue sky filtering which supposes that the clouds are separated (e.g., cumulus cloud field) and that they do not overlap on the image due to projection (this would result in merged contours).

**Reviewer #1:** In section 2.2, page 6 line 22, the authors mention the blur effect in the peripheral regions of the rectified image. To my understanding, the blur effect results from the mapping of a given image region of the fisheye image onto a larger projection area in the rectified image. I find the term "interpolation" misleading, because the rectification itself is performed in reverse, from given rectified image coordinates for which the corresponding coordinates in the fisheye image are computed. Since such an image mapping generally never hits the center of a pixel in the target image exactly, an interpolation is applied as a normal procedure (e.g bilinear).

→We modify the sentence to:
[…] In this transformation, the peripheral areas are mapped from a given region of the fisheye image onto a larger projection area in the rectified image, producing the blur effect.

**Reviewer #1:** There seems to be a minor error in equation 7, where the two vectors are written as row vectors, but have the transposition applied as in the normal co-linearity equation, which generally assumes column vectors. As far as i can see, the rest of the article uses column vectors (e.g. page 7 line 17). Hence, the row vector on the right side of the essential matrix must be a column vector and vice versa.

→ We thank the reviewer for catching this error. This mistake has been corrected.

**Reviewer #1:** In section 2.3, page 8 line 11, a constraint for the matrix R is introduced, which enforces R to be the identity. In general, the matrix R is to be computed in the relative orientation estimation procedure and should not be the identity matrix. Of course, the relative orientation of the cameras in a frontally aligned pose should be the identity, but not the matrix R, which is to be computed.

→The use of a mathematical formalism leads here to confusion and has been deleted. The paragraph has been modified.

"This validation is necessary because the algorithm is sensitive to the stereo pixel matching quality and can, in some cases, converge towards an incoherent minimum. In practice, we minimize $||T - (b, 0, 0)||^2 + ||R - Id||^2$ by varying subsets of matching pixels used in step 2 (E matrix computation)."

is re-written:

Consistency step: Initial visual orientation of the cameras is achieved to be as close as possible to the frontally aligned relative orientation (i.e., T= (b,0,0) and R = Id; see section 2.3). In our algorithm, several estimations of the essential matrix E, and consequently R and T, are achieved to avoid incorrect solutions which are due to erroneous or imprecise matches in the SIFT procedure. These estimations are

obtained by using several subsets of the matching pixel set given by the SIFT procedure. Estimations of E matrix, which are not coherent with the R ~ Id and T ~ (b,0,0) hypothesis are then rejected. Among the coherent estimations, we choose the one that leads to minimal corrective rotations.

**Reviewer #1:** In section 2.4, page 9 line 5, the authors describe a method to filter out outlier in the reconstruction by introducing a lower and upper limit of a valid height. This might work well for zenith regions, but for larger incidence angles mismatches introduce a larger error in the horizontal location rather than the vertical, even of the lower and upper limits for the height value are satisfied. In other words, a depth error moves from a vertical error to a more and more horizontal error as the incidence angle grows larger. This limitation should be mentioned here.

→We agree with this comment. The item page 9 line 5:

- adjusting algorithm parameters: height detection range can be limited during the pixel matching process by setting minimum and maximum bounds for cloud height detection ; window correlation size is adjusted to prevent speckles.

has been replaced by:

- adjusting algorithm parameters:
    - disparity range is limited during the pixel matching process by setting minimum and maximum bounds for cloud height detection. Note that disparity bounds can be related to height detection bounds with equation (Eq. 12), even if this relationship becomes less relevant for larger incident angles for which larger horizontal errors occur.

    - window correlation size is adjusted to prevent speckles.

**Reviewer #1:** Technical Corrections:
- Page 3 line 19+20: This sentence should be reformulated. For example: "The calibration of each camera encompasses a mathematical description of the projection of an incident optical ray onto the image."
- Page 10 line 2: "contour detection and segmentation using the binarized image: .."
- Page 10 line 18: "..., until a distance -w-here the quality..."
- Page 13 line 3: "undistorsion" - > "undistortion"
- Page 13 line 14: Should the delta_h be sigma_h?

→ These technical corrections have been addressed.

---

## Author Comment (AC2) · 24 Nov 2017

We thank the reviewer for his constructive comments and respond to him individually in the following text.

**Reviewer #2:** An additional limitation of the proposed setup might be in extracting vertical profiles of clouds, since cameras cannot see through clouds and all sky cameras work at zenith angle. The authors may consider discussing the impact of this limitation in investigating cloud's life cycle.

→ Extracting cloud vertical profiles can be achieved at large zenith angles as long as the cloud tops are not hidden in the projection. This has been shown by Beekmans et al. 2016 who compared such reconstructed profiles with cloud radar profiles with good agreement. However, cloud vertical extension parameter will be difficult to follow in a lagrangian way during a single cloud trajectory. Also, cloud life cycles can be followed simply following the projected area that increases and decreases with cloud formation and dissipation or converges into an overlying stratocumulus layer.

To discuss the impact of this limitation, the last paragraph of the article (page 15, line 2)

Finally, the use of photogrammetry techniques associated with segmentation opens the way to the characterization of other parameters of interest to the atmospheric science, such as the width of the cloud base and the vertical extension of the clouds, as shown by Beekmans et al. 2016. In addition, segmentation makes it possible to track individual clouds through successive images and follow the evolution of the cloud life cycle.

has been modified to:

Finally, the use of photogrammetry techniques associated with segmentation opens the way to the characterization of other parameters of interest in atmospheric science, such as the width of the cloud base and the vertical extension of the cloud. The width of cloud base follows its growth and dissipation, and can be well estimated at low zenith angles. In contrast, extracting cloud vertical dimensions can be achieved at large zenith angles as long as the cloud tops are not hidden in the projection (Beekmans et al. 2016). Consequently, segmentation makes it possible to track individual clouds through successive images and follow the evolution of the cloud life cycle by tracking cloud heights and/or cloud base widths.

**Reviewer #2:** Page 2, line 24, "an" needs to be "a" Table 3, replace "significative" with "significant"

→ These corrections have been addressed.

**Reviewer #2:** Figure 8/9, center/rightmost panel in the bottom row, are the numbers missing, what is "//"?

→ The symbol // was used to show that the ceilometer beam did not hit a cloud at zenith. It has been replaced by *"// No cloud"* for greater clarity.